# Microbial Signature in Adipose Tissue of Crohn’s Disease Patients

**DOI:** 10.3390/jcm9082448

**Published:** 2020-07-31

**Authors:** Carolina Serena, Maribel Queipo-Ortuño, Monica Millan, Lidia Sanchez-Alcoholado, Aleidis Caro, Beatriz Espina, Margarita Menacho, Michelle Bautista, Diandra Monfort-Ferré, Margarida Terrón-Puig, Catalina Núñez-Roa, Elsa Maymó-Masip, M. Mar Rodriguez, Francisco J. Tinahones, Eloy Espin, Marc Martí, Sonia Fernández-Veledo, Joan Vendrell

**Affiliations:** 1Hospital Universitari de Tarragona Joan XXIII, Institut d’Investigació Sanitària Pere Virgili Universitat Rovira i Virgili, 43005 Tarragona, Spain; carolina.serena@iispv.cat (C.S.); diandramonfort@gmail.com (D.M.-F.); margarida.terron@gmail.com (M.T.-P.); catinuroa@gmail.com (C.N.-R.); elsamaymomasip@gmail.com (E.M.-M.); m.marrodpe@gmail.com (M.M.R.); 2CIBER de Diabetes y Enfermedades Metabólicas Asociadas (CIBERDEM), Instituto de Salud Carlos III, 08029 Madrid, Spain; 3CIBER de Obesidad y Nutrición (CIBERObN), Instituto de Salud Carlos III, 08029 Madrid, Spain; maribelqo@gmail.com (M.Q.-O.); l.s.alcoholado@gmail.com (L.S.-A.); 4Unidad de Gestión Clínica Intercentros de Oncología Medica, Hospitales Universitarios Regional y Virgen de la Victoria de Málaga, Instituto de Investigación Biomédica de Málaga (IBIMA)-CIMES-UMA, 29010 Málaga, Spain; 5Colorectal Surgery Unit, Hospital Universitari Joan XXIII, 43007 Tarragona, Spain; monica.millan@ymail.com (M.M.); dra5028@gmail.com (A.C.); bespina84@gmail.com (B.E.); 6Colorectal Surgery Unit, Hospital Universitari La Fe, 46026 Valencia, Spain; 7Digestive Unit, University Hospital Joan XXIII, IISPV, 43007 Tarragona, Spain; mmenacho.hj23.ics@gencat.cat (M.M.); bmichelleannette@gmail.com (M.B.); 8Endocrinology and Nutrition Department, Biomedical Research Institute from Malaga (IBIMA), University Hospital Virgen de la Victoria of Malaga, Malaga University, 29010 Málaga, Spain; fjtinahones@uma.es; 9Colorectal Surgery Unit, General Surgery Service, Valle Hebron Hospital, Autonomous University of Barcelona, 08035 Barcelona, Spain; eespin@vhebron.net (E.E.); marcmartig@gmail.com (M.M.); 10Medicine and Surgery Department, School of Medicine, Universitat Rovira i Virgili, 43005 Tarragona, Spain

**Keywords:** tissue microbiota, inflammatory bowel disease, 16S sequencing, creeping fat, PICRUSt analysis, *Escherichia coli*, *Fusobacterium*, lipopolysaccharide biosynthesis

## Abstract

Crohn’s disease (CD) is characterized by compromised immune tolerance to the intestinal commensal microbiota, intestinal barrier inflammation, and hyperplasia of creeping fat (CF) and mesenteric adipose tissue (AT), which seems to be directly related to disease activity. Gut microbiota dysbiosis might be a determining factor in CD etiology, manifesting as a low microbial diversity and a high abundance of potentially pathogenic bacteria. We tested the hypothesis that CF is a reservoir of bacteria through 16S-rRNA sequencing of several AT depots of patients with active and inactive disease and controls. We found a microbiome signature within CF and mesenteric AT from patients, but not in subcutaneous fat. We failed to detect bacterial DNA in any fat depot of controls. Proteobacteria was the most abundant phylum in both CF and mesenteric AT, and positively correlated with fecal calprotectin/C-reactive protein. Notably, the clinical status of patients seemed to be related to the microbiome signature, as those with the inactive disease showed a reduction in the abundance of pathogenic bacteria. Predictive functional profiling revealed many metabolic pathways including lipopolysaccharide biosynthesis and sulfur metabolism overrepresented in active CD relative to that in inactive CD. Our findings demonstrate that microbiota dysbiosis associated with CD pathophysiology is reflected in AT and might contribute to disease severity.

## 1. Introduction

Inflammatory bowel diseases (IBDs) have become a worldwide epidemic in developed societies. They are a heterogeneous group of disorders of multifactorial etiology and are commonly represented by two major phenotypes—Crohn’s disease (CD) and ulcerative colitis (UC) —both characterized by persistent inflammation and ulcerations at the small or large bowel, a low rate of spontaneous remission, and an undulating course of activity with relapsing attacks after periods of remission. Dysbiosis and decreased complexity of the intestinal microbiota, owing to a shift in the balance between commensal and potentially pathogenic microorganisms, are well-recognized features in both CD and UC [1,2,3,4,5], with microbiota dysbiosis being significantly greater in CD than in UC [6]. Indeed, the gut microbiota is believed to be a driving factor for inflammation and the development of intestinal lesions, because the surgical diversion of the fecal stream has been shown to induce remission [7,8,9,10]. Mesenteric and intestinal disease manifestations are tightly coupled in CD and, in this context, some authors have recently claimed that the mesentery is a contiguous organ that may play an important role in immunological processes due to its anatomical position in the intestinal tract [11,12,13,14]. Hypertrophy of the mesenteric fat adjacent to the inflamed regions of the intestine, so-called “creeping fat (CF)” is a hallmark of CD and seems to be directly related to disease activity [15,16]. Indeed, it has been recently described that reoperation rates among patients with CD decrease dramatically, from 27% to 2.7%, when the mesentery is included during intestinal resection [13]. Mesenteric adipose tissue (AT) expansion in CD is mainly dependent on adipocyte hyperplasia, which occurs via recruitment and differentiation of AT precursors termed AT-derived stem cells (ASCs) [17,18]. ASCs not only participates in the turnover of mature adipocytes in humans, but they also possess immunoregulatory properties that can be induced by the underlying pathological state [19,20]. In this context, we recently demonstrated that ASCs isolated from patients with CD have augmented proliferative, invasive, and phagocytic capacities and seem to be key players in the development of CF [21].

Bacterial translocation and tissue microbiota in humans are subjects of intense debate since several years ago [18,22,23,24,25]. Although some authors hypothesize that adipose tissue microbiota may come from the intestine when leaky gut occurs, there is still no reliable evidence in CD. Thus, previous works identified the presence of bacterial translocation [18,26,27] instead of determining the adipose tissue microbiota profile, and none describes the evolutive changes according to the clinical status of CD patients.

In this study, we show that mesenteric AT is a bacterial reservoir and that patients with CD have a microbiome signature within mesenteric AT that is dependent on their clinical status. Accordingly, mesenteric AT might cooperate with the gut in a pro-inflammatory feedback loop, generating a specific microbiome signature in this tissue to influence CD status and/or disease progression.

## 2. Material and Methods

### 2.1. Study Design

Subjects were recruited at University Hospital Joan XXIII (Tarragona, Spain) and University Hospital Vall d´Hebrón (Barcelona, Spain) following the tenets of the Declaration of Helsinki. The corresponding hospital ethics committees approved the study, and written informed consent was obtained from all participants before entering the study. Donors were classified as those in relapse (active) or remission (inactive) following the Crohn’s Disease Activity Index criteria (CDAI score) as well as biological and clinical parameters such as PCR and fecal calprotectin. Furthermore, endoscopic evaluation was available in 75% of the patients, with a complete correspondence with the clinical classification obtained by CDAI [28,29].

Samples of mesenteric (*MES*-VAT) and subcutaneous (SAT) AT were obtained from patients with inactive CD undergoing non-acute surgical interventions such as hernia repair or cholecystectomy, in a scheduled routine surgery (*n* = 5). Moreover, samples of mesenteric pericolic “wrapping” AT (CF origin; *CF*-VAT), *MES*-VAT (away from the active lesion in a healthy mesentery), and SAT (*n* = 8) were obtained from patients with active or complicated CD submitted to surgery. Lastly, samples of *MES*-VAT and SAT were obtained from non-CD subjects (control group) undergoing non-acute surgical interventions such as hernia repair or cholecystectomy, in a scheduled routine surgery (*n* = 8). Tissue samples were aseptically collected and (when appropriate) all were obtained before bowel opening or resection to avoid possible contamination. Clinical data, anthropometric, and biochemical variables from the cohort are presented in Table 1.

### 2.2. RNA Extraction from Tissues

RNA was extracted from 200 mg of tissue using the TriPure Isolation Reagent (Roche, Basel, Switzerland). RNA concentration was determined by absorbance at 260 nm, and purity was estimated with a Nanodrop spectrophotometer (Nanodrop Technologies Inc., Wilmington, DE, UAS). cDNA was synthesized using SuperScript II reverse transcriptase and random hexamer primers (Invitrogen Life Technologies, Darmstadt, Germany).

### 2.3. PCR Amplification and Analysis of 16S-rRNA Sequences

For microbial community profiling, 16S-rRNA gene sequences were amplified from cDNA using the 16S Metagenomics Kit (Thermo Fisher Scientific, Madrid, Spain), which includes two primer sets of the 16S hypervariable regions in bacteria: primer set V2–4–8 and primer set V3–6, 7–9. PCR products were purified using Agencourt AMPure XP DNA purification beads (Beckman Coulter Genomics GmbH, Bernried, Germany) to remove primer dimers. The concentration and the average size of each amplicon were determined using the Quant-iT PicoGreen dsDNA Assay Kit (Invitrogen), and the amount of DNA fragments per microliter was calculated. Libraries were prepared using the Ion Plus Fragment Library Kit, and samples were labeled using the Ion Xpress Barcode Adapters 1–16 kit. Library concentrations were determined using the Ion Universal Library Quantification Kit. Emulsion PCR and sequencing of the amplicon libraries were carried out using the Ion Torrent S5 system and the Ion 520/530 Kit-Chef. After sequencing, the individual sequence reads were filtered using the Ion Reporter Software V4.0 to remove low-quality and polyclonal sequences. All Ion platforms and reagents were from Thermo Fischer Scientific.

### 2.4. Bioinformatic Analysis

Datasets were analyzed using Quantitative Insights into Microbial Ecology (QIIME II) 2017.10.0 software (http://qiime.org). Sequencing reads were de-multiplexed and further filtered through the split_library.py script of QIIME II. To guarantee a higher level of accuracy, the reads were excluded from analysis if they had an average quality score <25 and if there were ambiguous base calls. Operational taxonomic units (OTUs) were defined by clustering sequences at a similarity of >97% and the representative sequences, chosen as the most abundant in each cluster, were submitted to the UCLUST-consensus taxonomy assigner (https://drive5.com/usearch/manual/uclust_algo.html) to obtain the taxonomy assignment and the relative abundance of each OTU, using the Greengenes 16S rRNA gene database (http://greengenes.lbl.gov/cgi-bin/nph-index.cgi). α- and β-diversity were evaluated through QIIME II as described [30]. Distances between microbial communities from each sample were calculated with the jackknife coefficient; they were represented by the unweighted pair group method with arithmetic mean algorithm (UPGMA) clustering trees describing the dissimilarity between multiple samples generated by QIIME II. 16S gene sequences have been deposited in GenBank under different accession numbers.

We used PICRUSt analysis to predict metagenome function by picking OTUs against the Greengenes database, as previously described [31,32]. This OTU table was used for predicting metagenomes at different KEGG levels. The statistical analysis was performed in the R environment (v3.3.3). *p*-values were corrected for multiple testing using the Benjamini–Hochberg method [31].

### 2.5. Statistical Analysis

Statistical analysis was performed with the Statistical Package for the Social Sciences software, version 15 (SPSS, Chicago, IL, USA). For clinical and anthropometrical variables, normally distributed data were expressed as mean ± SD, and variables with no Gaussian distribution values were expressed as median (25th–75th quartiles). Student’s *t*-test with Bonferroni adjustment was used to compare the mean value of normally distributed continuous variables. For variables that did not have a Gaussian distribution, we used the Kruskal–Wallis test with post hoc Dunn’s multiple comparison test. To analyze the differences in nominal variables between groups, we used the χ2 test. For microbiota data, the statistical analysis was performed in R v3.3.3. Two-way analysis of variance (ANOVA), followed by Bonferroni’s test for multiple comparisons, was performed to check for significant differences. Pearson’s correlation coefficient with Bonferroni adjustment was used to analyze the relationship between parameters. Visualizations were performed with GraphPad Prime v6 (GraphPad Software Inc., San Diego, CA, USA).

### 2.6. Ethics Approval and Consent to Participate

The study has been carried out in accordance with the guidelines of the Declaration of Helsinki and the corresponding hospital ethics committees approved the study. Written informed consent was obtained from all participants before entering the study.

## 3. Results

### 3.1. Microbiome Signature of Mesenteric Adipose Tissue in Patients with Crohn’s Disease

Adipose tissue from creeping fat from active CD patients, showed a marked inflammatory infiltration with an increased expression of pro-inflammatory genes (Appendix A). Sequencing analysis of the 16S-rRNA gene sequences from these patients revealed a rich microbiome profile both in *CF*-VAT and in *MES*-VAT. By contrast, 16S-rRNA gene sequencing was negative in SAT depots of patients with CD (active and inactive) and in *MES*-VAT and SAT of non-CD subjects (controls) with the method employed in this study.

Weighted UniFrac principal coordinates analysis (PCoA), a measure of community composition, showed that the microbiome of *CF*-VAT and *MES*-VAT of patients with active CD had a similar pattern of clustering (*p* = 0.502), with the principal scores explaining 26%, 13% and 9% of the total variation (Figure 1A). No significant differences were found in α—diversity indices (Observed (*p* = 0.109), Chao1 (*p* = 0.115), ACE (*p* = 0.060), Shannon (*p* = 0.155), Simpson (*p* = 0.065), InvSimpson (*p* = 0.071), and Fisher (*p* = 0.193)) between the *CF*-VAT and *MES*-VAT microbiomes in patients with active CD (Figure 1B). However, we found significant differences in the relative abundance of several taxa between the *CF*-VAT and *MES*-VAT. Proteobacteria was the most abundant phylum in both *CF*-VAT and *MES*-VAT, followed by Bacteroidetes and Firmicutes. At the phylum level, Proteobacteria relative abundance was significantly higher in *CF*-VAT than in *MES*-VAT (70.64 ± 7.14% vs. 52.83 ± 9.25%, *p* = 0.00059) (Figure 1C). At the family level, the relative abundance of *Enterobacteriaceae* was significantly higher in *CF*-VAT than in *MES*-VAT (53.47 ± 7.16% vs. 40.29 ± 7.75%, *p* = 0.0091) (Figure 1D), whereas the relative abundance of *Bacteroidaceae* was significantly lower (10.93 ± 7.84% vs. 23.57 ± 7.03%, *p* = 0.0151).

At the genus level, *Bacteroides* relative abundance was significantly lower in *CF*-VAT than in *MES*-VAT (15.53 ± 8.84% vs. 31.59 ± 10.59%, *p* = 0.0234) (Figure 1E). Specifically, we found a lower relative abundance of the species *Bacteroides vulgatus* in *CF*-VAT relative to that in *MES*-VAT (1.12 ± 0.95% vs. 15.85 ± 6.96%, *p* = 0.0155) (Figure 1E).

### 3.2. The Microbiome Signature in Mesenteric Adipose Tissue in Crohn’s Disease Is Related to the Clinical Status

Next, we compared the microbiome of *MES*-VAT between patients with active and inactive disease. Weighted UniFrac PCoA showed a different cluster pattern for *MES*-VAT between active and inactive CD. The analysis of similarities with permutations revealed significant differences between active and inactive CD (*p* = 0.047), as demonstrated by the two principal component scores, which accounted for 32% and 21% of the total variation (Figure 2A). However, no significant differences were observed in α-diversity indices (Observed (*p* = 0.883), Chao1 (*p* = 0.194), ACE (*p* = 0.476), Shannon (*p* = 0.379), Simpson (*p* = 0.103), InvSimpson (*p* = 0.106), and Fisher (*p* = 0.707)) between the microbiome of active and inactive CD (Figure 2B). Nevertheless, taxonomy-based comparisons of *MES*-VAT at the phylum, family, and genus level showed significant differences between active and inactive CD. Of note, at the phylum level, we found a significant increase in the relative abundance of Proteobacteria (52.83 ± 9.98% vs. 30.39 ± 1.75%, *p* = 0.0302) and a significant decrease in the relative abundance of Bacteroidetes (17.49 ± 5.6% vs. 38.91 ± 2.5%, *p* = 0.0270) in active relative to inactive CD (Figure 2C). Additionally, we found a positive correlation between the relative abundance of Proteobacteria and fecal calprotectin (R = 0.68; *p* = 0.0005) and C-reactive protein (R = 0.418; *p* = 0.017) levels (Figure 2D).

At the family level, we found that the relative abundance of Enterobacteriaceae (40.29 ± 7.73% vs. 16.74 ± 2.12%, *p* < 0.0001) and *Fusobacteriaceae* (1.29 ± 0.73% vs. 0.03 ± 0.01%, *p* = 0.034) was significantly higher in active than in inactive CD, whereas the relative abundance of *Ruminococcaceae* was significantly lower (1.42 ± 0.64% vs. 2.78 ± 0.42%, *p* = 0.045) (Figure 2E). At the genus level, we found a significant increase in the relative abundance of *Escherichia* spp. (4.06 ± 0.9% vs. 0.54 ± 0.02%, *p* = 0.032) and *Fusobacterium* spp. (2.4 ± 0.74% vs. 0.04 ± 0.02%, *p* = 0.044) in active compared with inactive CD (Figure 2F). Lastly, at the species level, we found a significant decrease in the relative abundance of *Faecalibacterium prausnitzii* (0.20 ± 0.16% vs. 4.48 ± 0.24%, *p* = 0.039) and an increase in the relative abundance of *Escherichia coli* (7.46 ± 2.43% vs. 1.00 ± 0.18%, *p* = 0.0101) in the *MES*-VAT of patients with active CD compared with their inactive counterparts (Figure 2G).

### 3.3. Functional Differences in the Gut Microbiota in Mesenteric Adipose Tissue between Active and Inactive Crohn’s Disease

We used the PICRUSt tool to predict the function of the microbial communities based on the 16S datasets. No significant differences were detected at the functional level between the *CF*-VAT and *MES*-VAT depots in patients with active CD. Remarkably, however, we observed significant functional differences depending on the clinical status of the patient. Metabolism pathway genes (KEGG categories) for energy metabolism (*p* = 0.003) including oxidative phosphorylation (*p* = 0.011) and sulfur metabolism (*p* = 0.011), and also lipid metabolism pathways for fatty acid biosynthesis (*p* = 0.019) and secondary bile acid biosynthesis (*p* = 0.028), were all significantly over-represented in the *MES*-VAT of patients with active CD compared with their inactive counterparts. Likewise, the *MES*-VAT of patients with active CD showed significant enrichment in the proportion of genes related to lipopolysaccharide (LPS) biosynthesis (*p* = 0.006), polycyclic aromatic hydrocarbon degradation (*p* = 0.002), and cytochrome P450 (*p* = 0.003) relative to the *MES*-VAT of patients with inactive CD (Figure 3). Metagenomic comparison of the study groups showed that gene families linked to pathways in cancer (*p* = 0.002) and colorectal cancer (*p* = 0.045) were significantly increased in the *MES*-VAT of patients with active CD.

Finally, genes related to carbohydrate metabolism (*p* = 0.011) including glycolysis/gluconeogenesis (*p* = 0.011), pentose phosphate pathway (*p* = 0.030), fructose and mannose metabolism (*p* = 0.045), pentose and glucuronate interconversions (*p* = 0.030), linoleic acid metabolism (*p* = 0.030), and propanoate metabolism (*p* = 0.019), as well as genes for alanine, aspartate and glutamate metabolism (*p* = 0.045), nitrogen metabolism (*p* = 0.045) and mineral absorption (*p* = 0.011), were all significantly increased in the *MES*-VAT of patients with inactive CD relative to their active counterparts (Figure 3).

## 4. Discussion

In this study we report, for the first time to our knowledge, unique microbiome signatures in the VAT of patients with CD—both in *CF*-VAT and in *MES*-VAT. Moreover, our data indicate that clinical remission of CD does not preclude the presence of particular microbiota composition in mesenteric fat, pointing to this specific tissue as a reservoir for bacteria with the potential to be translocated across a disrupted intestinal barrier.

In recent years, advances in the molecular techniques used to characterize microbial communities have led to a greater appreciation of the diversity of microorganisms inhabiting several unexpected tissues of the human body, including internal tissues. In this line, a distinct diversity of bacteria has been recently described in different AT depots, including breast AT [33], compartments of skin AT [34], and heart AT [35]. Furthermore, the existence of a mesenteric AT microbiota was recently reported in mice using an optimized 16S metagenomics sequencing pipeline [36]. An earlier pyrosequencing study in stromal vascular cells from human AT [22] revealed a distinct microbiome in this depot and showed that the diversity of microbiota differed between lean, overweight, and obese persons. By contrast, other authors failed to find a human AT microbiota from obese or normal-weight subjects [23]. This latter observation is in line with the absence of microbial sequences in *MES*-VAT and SAT from healthy subjects in the present study. Of note, de Goffau and colleagues [24] recently demonstrated that the human placenta has no microbiome in a large prospective cohort using both metagenomics and 16S amplicon sequencing, which contrasts with what was previously reported [37,38,39]. The use of aseptic techniques and/or new-generation sequencing methods might explain these contrasting results. Remarkably, a very recent paper by Anhê and colleagues [25] provided evidence for a microbial profile in plasma and liver and three different AT depots (omental, *MES*-VAT and SAT) of individuals with morbid obesity (mean body mass index ~50 kg/cm^2^). Interestingly, they found that type 2 diabetes created an extra-intestinal microbial signature, independent of obesity, in the five compartments studied. Intriguingly, the authors observed a greater abundance of *Enterobacteriaceae* (specifically, *Escherichia* and *Shigella*) in the plasma and *MES*-VAT of subjects with type 2 diabetes. These data are in accord with our findings of *Enterobacteriaceae* as the most abundant family in the *MES*-VAT of patients with active CD. The aforementioned study thus uncovers a unique organ-specific microbial signature, or potential internal “tissue microbiota”, in obesity and type 2 diabetes [40].

CD and obesity have been proposed as diseases with similar characteristics in terms of manifestations in VAT. Indeed, alterations in body fat distribution and accumulation of intra-abdominal adipose tissue are well-recognized features in CD and obesity, which are associated with an increased prevalence of systemic inflammation and metabolic disturbances. Moreover, the disruption of the intestinal barrier resulting in a leaky gut has been claimed as a feature common to both diseases. However, whereas the mucosal barrier is only mildly affected in obesity, CD is characterized by severe transmural inflammation with subsequent destruction of the intestinal barrier. Bacteria penetrating the intestinal wall are likely to locate to the mesenteric fat, which could trigger adipocyte hyperplasia and provoke *CF*-VAT development or persistence [18]. This hypothesis is in line with the findings of our study, i.e., the discovery of bacteria within the *MES/CF*-VAT of patients with CD.

Of note, we found that the VAT microbiome of patients with CD is a mirror of gut dysbiosis. We detected a broad spectrum of microorganisms in VAT of patients with CD, both in *CF*-VAT and *MES*-VAT, with Proteobacteria and Bacteroidetes being the most abundant phyla. It has been described that the intestinal mucosa microbiota in CD differs greatly from that in UC or healthy subjects, with an increase in Bacteroidetes and Proteobacteria [41]. Remarkably, PCoA of the microbiome between *CF*-VAT and *MES*-VAT of patients with active CD revealed a similar pattern of clustering, indicating no differences in the microbiome between the fat depots. These findings are in agreement with the concept that considers the mesentery as a continuous organ [13,42,43]. Accordingly, *CF*-VAT and *MES*-VAT might serve as a bacterial reservoir. By contrast, PCoA of the *MES*-microbiome between patients with active and inactive disease showed a different cluster pattern, with an increase in *Enterobacteriaceae* such as *Escherichia* spp. or *Fusobacterium* spp. in the *MES*-VAT of patients with active disease. These microorganisms were previously demonstrated to play a role in the pathogenesis of IBD; specifically, the abundance of fecal Enterobacteriaceae bacteria is augmented both in patients and mice with IBD [44]. *E. coli*, particularly adherent-invasive *E. coli* (AIEC) strains, have been isolated from ileal CD biopsies [45] and are also found in patients with UC [46]. Additionally, mucosal samples show more pronounced enrichment than fecal samples [47]. This indicates that the inflammatory milieu in IBD may favor the growth of the Enterobacteriaceae clade. Of note, the anti-inflammatory drug mesalamine was found to attenuate intestinal inflammation and reduce the abundance of *Escherichia/Shigella* in IBD [48,49].

Fusobacteria is another clade of adherent and invasive bacteria that principally colonize the oral cavity and the gut, and are reported to be present at a higher abundance in the colonic mucosa of IBD patients than in healthy controls [50,51]. When administered by rectal enema in mice, *Fusobacterium varium* can cause colonic mucosal inflammation [52]. The invasive ability of *Fusobacterium* bacteria correlates positively with IBD severity in patients [53], indicating that invasive *Fusobacterium* species may impact IBD pathology. Furthermore, this species has been associated with colorectal cancer, which is the most frequent malignant complication in patients with IBD [54,55,56]. *Fusobacterium* spp. produce H_2_S through cysteine desulfhydrase activity, and we found relevant changes in gene abundance for sulfur metabolism depending on the clinical status of the patients. In agreement with our findings, some authors found enrichment for microbial taxa involved in sulfur metabolism, such as *Escherichia*, *Shigella,* and *Fusobacterium*, in fecal samples of patients with active CD [57,58].

There are also specific groups of gut bacteria that might play a protective role against IBD. A range of bacterial species, most notably the genera *Lactobacillus*, *Bifidobacterium,* and *Faecalibacterium*, have been shown to protect the host from mucosal inflammation via several mechanisms, including the stimulation of anti-inflammatory cytokines [59] including IL-10, and the down-regulation of inflammatory cytokines [60]. In this context, our study reveals an increase (although not significant) in the abundance of both *Bifidobacterium* and *Faecalibacterium* in patients with inactive relative to active disease. These findings may indicate an increase in beneficial bacteria in patients with remission of the disease. This is consistent with the observations that patients with CD and a low abundance of *F. prausnitzii* in the mucosa are more likely to relapse after surgery [59], whereas restoration of *F. prausnitzii* after recurrence is associated with maintenance of clinical remission of UC [61]. Of note, we found a significant decrease in *F. prausnitzii* in the *MES*-VAT depot of active patients relative to those in remission of the disease. The *MES*-VAT of patients with active CD also showed a trend for an increase in the abundance of the genus *Streptococcus* compared with the *MES*-VAT of patients with inactive CD. Interestingly, the presence of *Streptococcus* spp. in stool samples before surgery is a predictive marker of future recurrence [6].

In our cohort, analysis of the inferred metagenome showed no significant differences between *CF*-VAT and *MES*-VAT of patients with active CD. This finding may indicate that both fat depots, which are contiguous, have a shared tissue microbiota. However, inferred functional analysis of microbiota in VAT from patients with active disease showed enrichment in the abundance of genes involved in LPS biosynthesis relative to patients in remission. LPS is an important component of the outer membrane of Gram-negative bacteria and plays a key role in triggering inflammatory responses in various diseases such as CD. Following our findings, a recent study [62] found that the LPS biosynthesis pathway in mucosal samples of patients with CD was significantly reduced in disease remission, whereas the abundance of pathways involved in glycolysis/gluconeogenesis and starch and sucrose metabolism was increased, suggesting that the induction of remission could partially rectify the gut microbiota dysbiosis and restore metabolic homeostasis.

Our data also showed an increase in genes for nitrogen metabolism; propanoate metabolism; and alanine, aspartate, and glutamate metabolism, in the same line as that found by He and colleagues [62]. Overall, these data indicate that the remission of the disease modifies the microbiota and its metabolic functional patterns in *MES*-VAT.

Intriguingly, the PICRUSt analysis revealed differences in the abundance of gene families linked to cancer in the samples of active CD patients, which might indicate the genomic potential of the microbes present in *MES*-VAT of active patients to develop colorectal cancer in these patients. In fact, in this study, we have found in *MES*-VAT of active patients a significant increase in the abundance of *Fusobacterium* spp. and Enterobacteriaceae (*Escherichia coli*), which as previously described may contribute to colorectal carcinogenesis [63,64,65]. Indeed, several recent studies have shown associations between colorectal cancer and CD activity [66,67,68] although a direct cause has not yet been demonstrated. The implication of gut microbiota in the link between CD and cancer would be immense, particularly for microbiota-based therapeutics. Nevertheless, we acknowledge the limitation of gene functional profiles inferred from 16S sequences, which are predictions only and we interpret these results with caution. Subsequent in-depth studies by whole-genome methods, such as whole community shotgun sequencing and RNA-seq are necessary to increase the insights into the relationships between the functionality of the *MES*-VAT microbiome and carcinogenesis in active CD patients. Even though *MES*-VAT from active vs. remission CD patients comes from different intraabdominal localizations, all are representative of visceral fat, and the influence on the results is probably low, however, it must be taken into consideration when interpreting the results. Furthermore, we are aware of the low number of subjects included in this study, and the similar immunosuppressive therapies in both CD groups, that may have influenced the microbial signature, nonetheless, changes will be induced in the same sense in both CD groups. All of this warrant further studies in larger cohorts to confirm these results.

In summary, our study points to VAT as a potential barrier against escaped gut bacteria in patients with CD, which might drive inflammation in this tissue. We found a microbiome signature enriched in Proteobacteria in the *CF*-VAT and *MES*-VAT depots of patients with CD associated with gut dysbiosis. Interestingly, the clinical status of patients altered the abundance of bacteria in VAT, and patients with inactive disease showed a lower abundance of potentially pathogenic bacteria (i.e., Proteobacteria) and a higher abundance of common mucosal bacteria (i.e., Bacteroidetes), as well as a lower abundance of genes involved in LPS biosynthesis compared with patients with active disease. Further studies are needed to determine the pathophysiological role of bacteria (or bacterial DNA) in AT of patients with CD.

## 5. Conclusions

Using 16S-rRNA-based approaches we found for the first time (to our knowledge) that mesenteric adipose tissue is a bacterial reservoir in patients with Crohn’s disease. Our data indicate that clinical remission of Crohn’s disease does not preclude the presence of a specific microbiota composition in mesenteric fat, pointing to this unique tissue as a well of bacteria with the potential to be translocated across a disrupted intestinal barrier. These novel data revealing a specific microbiome signature in adipose tissue from patients, with a strong relationship with disease activity, might provide the foundations for a new perspective of Crohn’s disease that may help to better manage patients.

### List of Abbreviations

CD, Crohn’s disease; CF, creeping fat; ASCs, adipose-tissue stem cells; IBD, inflammatory bowel disease; UC, ulcerative colitis; AT, adipose tissue; SAT, subcutaneous adipose tissue; VAT, visceral adipose tissue; MES, mesentery adipose tissue; BMI, body mass index; QUIME II, Quantitative Insights into Microbial Ecology; OTU, operational taxonomic unit; SPSS, Statistical Package for the Social Sciences software; SD, standard deviation; PCoA, Weighted UniFrac principal coordinates analysis; AIEC, adherent-invasive *E. coli* strains; LPS, lipopolysaccharide.

## Figures and Tables

**Figure 1 jcm-09-02448-f001:**
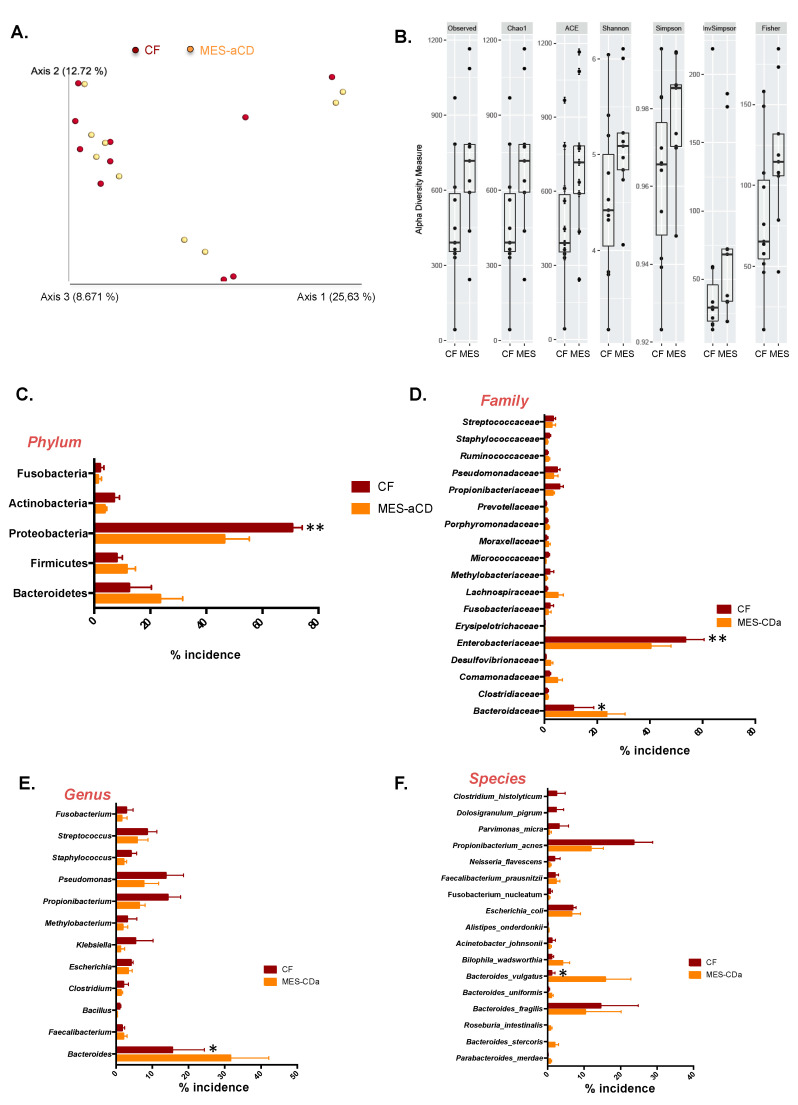
Visceral adipose tissue of patients with active Crohn’s disease contains bacterial DNA. (**A**) Weighted UniFrac principal coordinates analysis of microbiome data from creeping fat-visceral adipose tissue (*CF*-VAT) and mesenteric (*MES*)-VAT from eight patients with active Crohn’s disease (CD) revealed a similar pattern of clustering. (**B**) α-diversity indices (Observed, Chao1, ACE, Shannon, Simpson, InvSimpson, and Fisher) of the microbiomes of *CF*-VAT and *MES*-VAT. (**C**) Percentage of incidence of phylum, (**D**) family, (**E**) genus, and (**F**) species levels between *CF*-VAT and *MES*-VAT of CD patients. Data information: for B, data are represented in box-and-whisker plot format (whiskers: min to max). For C to F, values are expressed as mean ± SEM. Statistical analyses: Two-way analysis of variance with post hoc Bonferroni multiple comparisons test. *, p < 0.05; **, p < 0.01.

**Figure 2 jcm-09-02448-f002:**
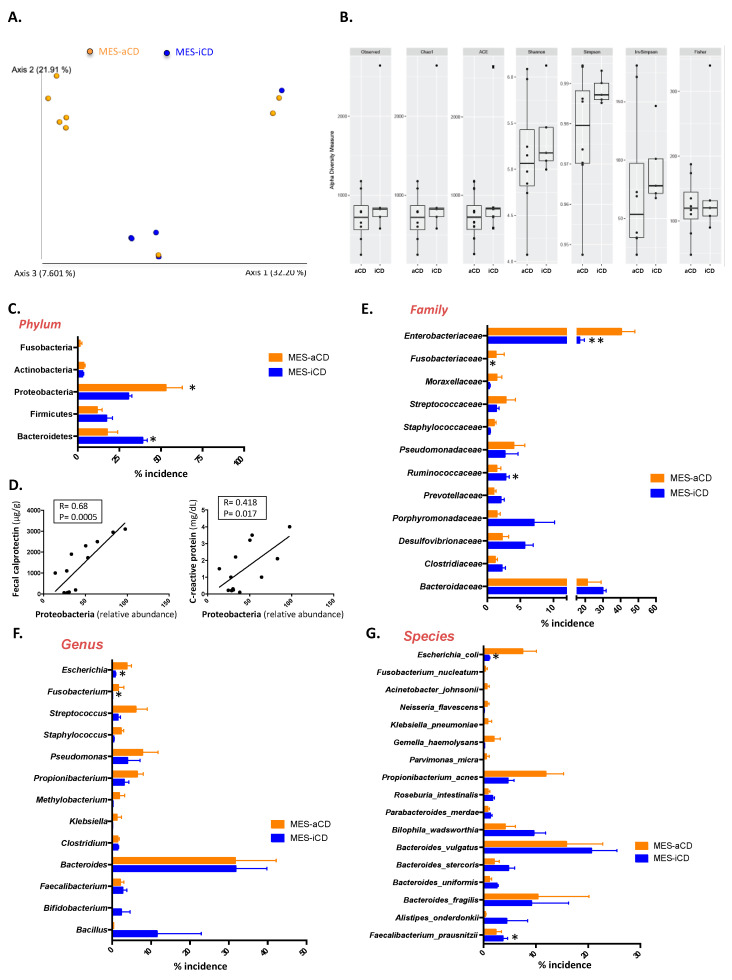
Distinct microbiome signatures in visceral adipose tissue of patients with active/inactive Crohn’s disease. (**A**) Weighted UniFrac principal coordinates analysis of microbiome data from mesenteric-visceral adipose tissue (*MES*-VAT) of patients with active or inactive Crohn’s disease (CD) shows two distinct clusters. (**B**) α-diversity indices (Observed, Chao1, ACE, Shannon, Simpson, InvSimpson, and Fisher) of the microbiome of *MES*-VAT of patients with active or inactive disease. (**C**) Percentage of the abundance of phylum levels between *MES*-VAT of active and inactive patients. (**D**) Positive correlation between Proteobacteria (relative abundance) and fecal calprotectin or C-reactive protein. Percentage of incidence in (**E**) family, (**F**) genus, and (**G**) species taxon levels in *MES*-VAT of patients with active or inactive disease. Data information: for B, data are represented in box-and-whisker plot format (whiskers: min to max). For C and E to G, values are expressed as mean ± SEM. Statistical analyses: Two-way analysis of variance with post hoc Bonferroni multiple comparisons test. For D, Pearson’s correlation analysis with Bonferroni adjustment was used. *, *p* < 0.05; **, *p* < 0.001.

**Figure 3 jcm-09-02448-f003:**
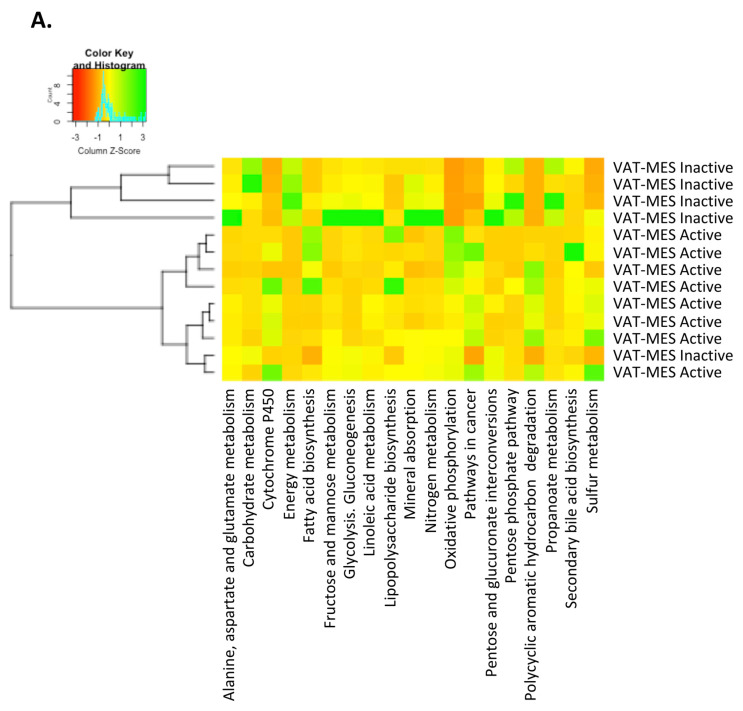
The inferred functional content of gut microbiota in visceral adipose tissue depends on the clinical status of the patients. (**A**) Heat map of functional content in mesenteric-visceral adipose tissue from patients with active and inactive Crohn’s disease was predicted by inferred PICRUSt analysis. *p*-values are corrected for multiple comparisons using the Benjamini–Hochberg method.

**Table 1 jcm-09-02448-t001:** Demographic and clinical characteristics of the cohort.

	Control	Inactive CD	Active CD
***n***	8	5	8
Sex (male/female)	4/4	2/3	4/4
Age (years)	46.1 (37.2–55.1)	47.6 (35.2–60.1)	45.8 (35.4–58.5)
BMI (kg/m^2^)	25.8 (20.2–29.1)	21.59 (20.1–24.1)	23.8 (20.5–27.3)
Glucose (mg/dL)	96.6 ± 6.3	94.6 ± 5.4	84.83 ± 4.7^a^
Smoking, *n* (%)	4 (50)	3 (60)	4 (50)
Cholesterol (mg/dL)	131 ± 11.2	118.2 ± 10.9	119.2 ± 15
HDL (mg/dL)	30.7 ± 4.5	31.13 ± 3.2	31.4 ± 6.4
Triglycerides (mg/dL)	186 (167.2–205.5)	186 (170.3–199.2)	119.6 (100.9–140) ^a,b^
Insulin (µIU/mL)	2.05 ± 1.9	4.98 ± 1.1 ^a^	6.12 ± 2.3 ^a^
HOMA-IR	0.7 ± 0.4	1.18 ± 0.5	1.44 ± 0.7 ^a,b^
Age at diagnosis (years)		26.5 ± 6.6	28.1 ± 8.5
Time in remission (months)		29 ± 9.3	
Indication of surgery	CoH	CoH	5SCD/5FCD
Immunomodulator use		5/5	10/10
Biological agent treatment		2/5	4/10
Steroid treatment		4/5	7/10
C-reactive protein (mg/dL)	0.05 ± 0.04	0.3 ± 0.13 ^a^	3.4 ± 1.6 ^a,b^
Fecal calprotectin (µg/g)		81.25 ± 20.3	2132.4 ± 159.1 ^b^

Abbreviations: BMI, body mass index; HDLc, high-density lipoprotein cholesterol; CoH, cholecystectomy or hernia; CD; Crohn’s disease; SCD, stenotic Crohn’s disease; FCD, fistulizing Crohn’s disease; HOMA-IR, Homeostatic Model Assessment for Insulin Resistance. Results are presented as mean ± SD or median (25th–75th percentiles), as appropriate. Differences were analyzed as described in Material and Methods. ^a^, *p* < 0.01 versus control group (non-CD subjects). ^b^, *p* < 0.01 versus inactive CD subjects.

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
