# Peer review of "Microbial Signature in Adipose Tissue of Crohn’s Disease Patients"

_jcm, 2020, doi:10.3390/jcm9082448_

Round 1

Reviewer 1 Report

This is very interesting study originally reporting unique microbiome signatures in the visceral adipose tissue of patients with Crohn's Disease. Of importance the presented data indicate that clinical remission in patients with Crohn's Disease does not preclude the presence of a particular microbiota composition in mesenteric fat. It is of clinical relevance as fat tissue might be an important but not widely recognized reservoir for bacteria with the potential to be translocated across the intestinal barrier, which might be a source of disease relapse. The paper is easy to follow, well structured. The graphs perhaps could be of better quality and bigger. As only 8 patients were included, however followed by sophisticated molecular analysis - this project could be viewed as pilot and the results should be verified in larger study. Otherwise I would recommend this paper for publication in JCM. 

Author Response

Thank you very much for your comments. We have changed the figures to increase quality.

We strongly agree with the reviewer that this study must be considered a pilot study and further studies with larger cohorts are necessary to verify this new data. We have included a comment regarding this limitation in the discussion section in the manuscript (page 17, lines 426-430).

Reviewer 2 Report

The study was excellent.

The purpose, planning, methods, results, and discussion were reasonable. References are appropriate.

This paper needs some histological figures, including inflammatory changes for the clear image of "creeping fat".

Author Response

Thank you very much for your comments.

We have included a new supplementary figure S1 with a representative histological image of the creeping fat from an active disease subject, and an image reflecting the inflammatory pattern from a molecular point of view of this tissue when compared with a healthy adipose tissue sample from the same depot (see supplementary figure S1 and Results section page 8, lines 202-204).

Reviewer 3 Report

This manuscript is of significant interest to the IBD community and addresses a novel concept- that of bacterial translocation to a novel compartment, visceral and mesenteric adipose tissue- in Crohn's disease.  The authors were able to detect a microbiome signature in creeping fat and mesenteric AT of patients with active Crohn's disease and within the mesenteric AT of patients with established but inactive Crohn's patients but not in controls and also in none of these patients within subcutaneous fat, which is quite fascinating and biologically plausible. Furthermore, the detected changes in microbiota are consistent with those expected based on dysbiosis in Crohn's disease and active IBD. The findings may help to explain disease pathogenesis and symptoms, as well as possible links for IBD-associated cancers. The manuscript is clearly written and follows a logical presentation. Please find minor criticisms below. Please note, my ability to critique the microbiota and computational methodology is limited, but these methods do seem appropriate to address the questions in the manuscript. 

  1. The use of CDAI to assess active disease is somewhat dated, as compared to more objective endoscopic and biochemical data or radiologic data. This is a limitation of the methodology. However, in the demographic table, it is nicely demonstrated that the active group has a significantly higher CRP and calprotectin than the inactive IBD group which does suggest accurate labeling. Addressing this point in the text would be important to highlight that the two groups are truly different.
  2. Congratulations to the authors for identifying a way to obtain mesenteric fat from patients with IBD who did not require abdominal surgeries.  This is a major challenge of such work and ultimately limits sample acquisition for translational studies. While I believe harvesting fat during hernia repair, etc, is the best method available, could differences in surgery type impact your findings at all? I do see that fat was harvested prior to enterotomy in the active and sterile techniques were used so this will likely minimize the likelihood of contamination.  Perhaps the difference in type of surgery can be discussed in the limitations briefly. 
  3. All patients in the active group were under treatment with biologics or immune modulators, and most were on steroids. A good number (about half) in the inactive group were also on similar therapies and so given the low disease activity this may suggest some mis-labeling of the inactive patients first and second could signify that some of the differences between Crohns' and non Crohn's patients might conceivably be attributed to medications and not disease related characteristics. This is unlikely but could be discussed as a limitation
  4. In the discussion, the active group is compared to inactive with the use of increased xxxx as compared to inactive. Later (paragraph starting with line 331) it is listed as Increased relative abundance of Lacto, bidifo, and fecalo in inactive as compared to active. This may be confusing to the reader. I suggest keeping the directionality the same and stating Lacto etc were DECREASED in active as compared to inactive. 

Author Response

We are very grateful for the comments of the reviewer. Please, find below the responses to the queries raised by the reviewer.

  1. We fully agree with the reviewer that the use of CDAI to assess active disease is risky compared to colonic endoscopic evaluation. Indeed, we would like to note that the classification of the patients in the former version was assessed by the CDAI score as well as biological and clinical parameters such as PCR and fecal calprotectin. However, complete definition of remission indeed depends on clinical and biological criteria including colon endoscopic evaluation. We would like to note that colon endoscopic evaluation was only available in 75% of patients. Nonetheless, the former classification of the subjects according to disease activity (following CDAI score as well as clinical and biological parameters) matches with “endoscopic remission” when available. We have included this information in the manuscript (Page 5, lines 121-124).
  2. Obtaining a sample of adipose tissue from Crohn’s disease patients in a safe and enough sterile condition to be analysed for microbiota studies, was one of the main challenges we faced in this study. This was especially difficult in subjects with remission disease because of the coordination between surgeons and gastroenterologists to identify patients scheduled for a minor surgical procedure, such as an inguinal hernia. Furthermore, to strength the sterility measures obtaining the sample of adipose tissue before any other surgical technical procedure was one of the main concerns that we had to overcome.

    Regarding the observations issued by the reviewer, we have clarified the localization of “healthy” mesenteric adipose tissue obtained from active disease patients (see lines 128-129, page 5 from study design). Likewise, we agree that the different origin of the VAT depot analysed in inactive CD patients could have shown differences due to the origin of the tissue however all samples are representative of visceral fat, even though they come from healthy mesentery away from to creeping fat in active CD versus visceral omentum from hernias or cholecystectomies in remission CD patients. We have included a comment about the adipose tissue localization in the discussion section (lines 423-430, page 17).

  3. The reviewer raises an excellent point. In this sense, we are now working on the effect of immunosuppressive therapy on immune-competent cells from adipose tissue in these patients to address some of the effects produced by these medications. As it is well known, it is very difficult to have non-active patients free of any pharmacological treatment, and on the opposite, to have active CD patients without immunomodulators. We agree with the reviewer observation, but the existence of patients on immunosuppressive therapy in both groups, in case of having influenced the results, they would have occurred in the same direction. We have included a comment on the discussion section (line 427, page 17).

  4. Thank you for your suggestion. We have rephrased the sentence (page 15, lines 385-386).